# sncRNAs in Epididymosomes: The Contribution to Embryonic Development and Offspring Health

**DOI:** 10.3390/ijms231810851

**Published:** 2022-09-16

**Authors:** Jingwen Luo, Xia Tan, Honggang Li, Xiaofang Ding

**Affiliations:** 1Center of Reproductive Medicine, Union Hospital, Tongji Medical College, Huazhong University of Science and Technology, Wuhan 430022, China; 2Institute of Reproductive Health, Tongji Medical College, Huazhong University of Science and Technology, Wuhan 430022, China

**Keywords:** epididymosomes, environment, sncRANs, embryonic development, offspring health, epigenetic

## Abstract

Much progress has been made in determining that paternal environmental exposures can remodel their spermatozoa small noncoding RNAs (sncRANs) and, in turn, affect the phenotypes of their offspring. Studies have shown that changes in the spermatozoa sncRNAs profile occur during passing through the epididymis. Due to the absence of transcription and translation in the epididymis, spermatozoa remodel their sncRNAs profile through communication with the epididymal microenvironment. Since epididymosomes contribute to the process of spermatozoa maturation by mediating the crosstalk between the epididymis and the passing spermatozoa, they are considered to be the leading candidate to mediate these changes. Previous studies and reviews on the role of epididymal transfer proteins in sperm maturation and function are myriad. This review focuses on the role and mechanisms of epididymosome-mediated transfer of sncRNAs cargoes onembryonic development and offspring health.

## 1. Introduction

Extracellular vesicles (EVs) are spherical particles encapsulated by phospholipid bilayers that are released by cells throughout the body [1]. They have received increasing attention and have become a hotspot in the field of medical research due to their functions of participating in intercellular communication, regulating physiological and pathological processes, as well as being used as diagnostic markers of diseases. EVs can be classified into three major subtypes according to their sizes, biogenesis, and functions: exosomes, microvesicles, and apoptotic bodies [2]. Epididymosomes are exosomes released by the epididymal epithelium, which are key elements of the epididymal microenvironment [3].

The fertilizing capacity of spermatozoa is dependent upon a wide range of biological properties, including competent viability, forward motility, as well as the ability to complete the acrosome reaction, zona pellucida reaction, and so on. However, spermatozoa are thought to be transcriptionally and translationally inert [3]; they cannot acquire morphological and functional maturity until undergoing a complex series of biochemical modifications in the epididymis. Therefore, studies on epididymosomes have focused on their role in sperm maturation for a long time. A large number of experiments have confirmed that the transfer of cargoes from epididymosomes to the developing spermatozoa is a primary mechanism for the establishment of sperm competency [4], such as promoting sperm forward progressive motility [5], participating in sperm capacitation [6,7], and modulating sperm–oocyte interaction [7].

Recent studies have suggested that epididymosome-transmitted sncRANs may go beyond affecting sperm function and extend to be important for embryonic development and offspring health [8,9,10]. This is a great step forward in exploring the contribution of epididymosomes in male reproduction.

What is more, new investigations are challenging previous studies. As the sperm and oocyte fuse to generate a zygote, the orthodox view is that genetic information can only be transmitted from gametes to somatic cells. However, according to Darwin’s pangenesis hypothesis, all cells in the paternal body can release “Gemmules”, which will pass the Weismann barrier and transmit information to spermatozoa. As a result, they can participate in the generation of offspring and potentially cause ancestral acquired features to reappear in succeeding generations [11]. This hypothesis has been ignored for hundreds of years. However, with the discovery of circulating nucleic acid and EVs, people began to appreciate its research significance and believed that the epididymis plays a crucial role in it. In the mirror of this hypothesis, the paternal circulatory nanovesicles, which are altered by environmental factors, can contact and exchange contents with epididymosomes to modify immature spermatozoa and mediate the transmission of paternal acquired characteristics [12]. The known functions of epididymosomes are quite analogous to the “Gemmules” proposed by Darwin [11]. Therefore, further exploration of epididymosomes may explain these genetic hypotheses.

Understanding the role of epididymosome-related scnRNAs inembryonic development and offspring health will not only help us better understand how parental environmental exposures affect offspring phenotypes, but it will also reduce the damage to offspring health.

## 2. Epididymosomes

The epididymal epithelium is mainly composed ofprincipal cells, apical cells, narrow cells, clear cells, basal cells, and halo cells [13].Epididymosomes are mainly produced and released by principal cells [14]. Besides, recent studies have demonstrated that clear cells can also produce epididymosomes [15]. Anatomically, the epididymis can be divided into three regions: caput, corpus, and cauda [16]. The heterogeneity of epididymosomes in each region coincides with the feature that each segment of the epididymis has unique characteristics and functions, suggesting that epididymosomes are closely related to epididymal function.

Epididymosomes are released in an apocrine secretory manner. This mechanism is characterized by the formation of cytoplasmic vesicles on the surface of the secretory epithelium of the epididymis. Following detachment, they enter and release their contents into the luminal environment [3].

Epididymosomes contribute to sperm maturation by mediating the crosstalk between the epididymis and spermatozoa [6]. Previous studies have shown that epididymosome-mediated proteins transfer promotes sperm forward motility [5], regulates sperm vitality [17], protects sperm from damage [18], removes misshapen and defective sperm [19], as well as participates in the processes of sperm–egg plasma membrane binding [7] and the zona pellucida reaction [6].

## 3. The Mechanisms of Epididymosomal Cargoes Transfer to Spermatozoa

Epididymosomes preferentially interact with the post-acrosomal sheath, which is located in the posterior region of the sperm head [3]. Since spermatozoa lack an endocytosis mechanism as well as organelles such as lysosomes [20], the exact mechanism of the binding of epididymosomes to spermatozoa has not been elucidated. The possible mechanism is briefly described here.

### 3.1. Delivery of Cargoes through Transient Fusion Pores

Epididymosomes are tightly bound to the spermatozoa via the receptor-ligand. Several membrane transport proteins have been identified in the proteome of epididymosomes to support this conjecture, such as SNARE proteins [20], Ras-like proteins [20], RAB small GTPase [3], etc. Following their combination, transient and incomplete membrane fusion ensues [21]. Additionally, once the delivery of the contents is complete, some epididymosomes immediately separate from the spermatozoa and re-enter the epididymal lumen [20]. It can also explain why a portion of epididymosomes persists in seminal plasma without being absorbed by spermatozoa.

### 3.2. Lipid Raft-Mediated Cargoes Transfer

Spermatozoa and epididymosomes’ membranes have regions rich in sphingomyelin and cholesterol, which are termed lipid rafts [7]. Some epididymal proteins, such as P25b [18] and SPAM1 [20], are anchored to lipid rafts in specific surface regions of spermatozoa via glycosylphosphatidylinositol (GPI) [3,7]. Thus, they can deliver cargoes to the spermatozoa. Besides, lipid rafts are associated with the relocation of DNM1 (dynamin1), which is involved in epididymosomes-mediated protein transfer to spermatozoa [21]. What is more, it is reported that the MFGE8 integrin-binding Arg-Gly-Asp (RGD) tripeptide motif plays an important role in the binding and cargo exchange between epididymosomes and spermatozoa [22]. RGD binding to αV integrin stimulates intracellular signaling pathways that support the recruitment of lipid rafts and their associated fusion mechanisms [22].

## 4. The Effects of Spermatozoa sncRANs in Embryonic Development and Offspring Health

### 4.1. The Role of Spermatozoa sncRANs in Embryonic Development

Spermatozoa sncRANs were initially assumed to be remnants of the spermatogenesis process. However, with the advancement of modern technology, they have been observed not to be static, but to exhibit considerable plasticity in the epididymis [23]. Specifically, the microRNAs (miRNAs) class of the spermatozoa sncRANs’ landscape declines dramatically, from 50% in caput to 16% in cauda spermatozoa [24]. Transfer RNA-derived small RNAs (tsRNAs), on the other hand, are only minutely abundant in the caput spermatozoa but are the most dominant classes in the cauda epididymis [24]. Concurrently, the abundance of other classes of sncRANs also fluctuates with the spermatozoa navigating the epididymis. When compared to the caput, the cauda spermatozoa show an increased abundance of piRNAs (Piwi-interacting RNAs) [25], while showing a decreased abundance of Small nucleolar RNAs (snoRNAs) and Small nuclear ribonucleic acids (snRNAs) [24]. Besides, the composition of each sncRAN class also undergoes constant remodeling. When spermatozoa miRNAs from distinct segments of the epididymis are sequenced and analyzed separately, some miRNAs, such as miR-471-5p, are characterized by a >256-fold increase in abundance between the caput and cauda epididymis. Conversely, approximately 24% of miRNAs are significantly lost during spermatozoa transit to the cauda [23]. However, far less is known about why sncRNAs undergo various alterations during the spermatozoa navigating the epididymis. Furthermore, it is apparent that more research is needed on whether altered spermatozoa sncRNAs, which play an important role in the maturation of sperm function, also have an impact on embryonic development.

Conine et al. generated zygotes via intracytoplasmic sperm injection (ICSI) using spermatozoa collected from the testis, caput and cauda epididymis [26]. Multiple epigenetic regulators were found to be overexpressed in embryos generated using caput spermatozoa. Furthermore, these embryos either failed to implant or perished soon afterwards. In contrast, embryos generated using testicular and cauda spermatozoa could successfully develop to term. Furthermore, they demonstrated that during epididymal transit, spermatozoa acquired miRNAs or sncRANs similar in size to them, rather than the class of longer tRNA fragments. They subsequently microinjected purified sncRNAs isolated from cauda spermatozoa into caput-derived embryos and found that pre-implantation molecular defects were rescued and that post-implantation survival rate was improved. Taking all these factors into consideration, they naturally concluded that remodeling of spermatozoa sncRANs during epididymal transit exerts an important function in embryonic development [26]. Contrasting observations had also been made, however, where caput spermatozoa could successfully generate healthy embryos via ICSI [27,28,29]. The reason for this discrepancy may be that the mechanical shearing of the needle used by Conine et al. damaged the sncRANs in the caput spermatozoa [30].

Even considering inconsistencies in the results of these experiments, they have extensively fueled interest in the study of spermatozoa sncRNAs. Up to now, many high-quality studies have been conducted on the subject of spermatozoa sncRNAs, especially miRNAs and tsRNAs, affecting embryonic development.

MiRNAs are a class of small non-coding RNAs, which can base pair with target gene mRNA and repress gene expression by directing the RISC (RNA-induced silencing complex) to degrade mRNA [31]. They are not only present in cells, but are also abundantly contained in EVs released by cells [12]. It has been detected that miRNAs’ profile changes dramatically from zygotes to blastocysts [32]. Some miRNAs that are essential for embryonic development, such as miR-34c, which is required for the first cell division of mouse embryos, can be detected in spermatozoa and zygotes, but not in oocytes [33]. When Dicer and Drosha (two miRNAs processing enzymes essential for miRNAs’ maturation) are selectively knocked out, the spermatozoa can still fertilize but generate embryos with decreased developmental potential, which can be rescued by injecting purified spermatozoa RNA [34]. Clinically, spermatozoa miR-191-5p has been proven to increase positive health outcomes in children conceived with Assisted Reproductive Technology (ART) [35].

TsRNAs processed from the mature tRNA or precursor tRNA are similar in abundance to miRNAs but are evolutionarily more conserved [36]. Previous studies have profiled the tsRNAs’ population in spermatozoa [37], epididymosomes [38], and embryos [39]. Remarkably, there are differences in tsRNAs between high- and low-quality embryos [40], implying that tsRNAs could potentially be used as biomarkers of embryo quality. Furthermore, this discrepancy triggers the question of whether poor embryo quality is due to defective embryo development caused by the abnormal expression of tsRNAs. Intriguingly, it has been demonstrated that sperm tRNA-Gln-TTG-derived small RNAs (Gln-TTGs) can interact with proteins such as HNRNPA2B1 and IGF2BP3 to participate in embryo cleavage and also regulate non-coding RNA processing to impact the expression of transcripts for embryonic genome activation [41]. Thus, further research on the role of tsRNAs in embryonic development could be an informative avenue.

In conclusion, sncRANs can be delivered to the oocyte along with the spermatozoa, in addition to being responsible for sperm maturation [6]. Recent evidence indicates that spermatozoa sncRANs are essential for processes such as embryonic cell proliferation and differentiation, as well as embryo implantation. Therefore, additional insight is needed, not only into the way spermatozoa RNA enters the oocyte, but also on a broader scale to understand the specific mechanisms of how spermatozoa sncRNAs affect embryonic development. Determining the possible roles of spermatozoa sncRNAs at various stages of embryo development may assist in rescuing embryonic defects, improving embryo quality, and increasing embryo fertilization rates.

### 4.2. The Role of Spermatozoa sncRNAs in Offspring Health

The impact of parental lifestyle on the health of offspring is now widely recognized. Male fetuses from fathers fed high-fat diets have significantly smaller placentas and lower birth weights [42]. In adulthood, their reproductive capacity declines [43]. Similarly, paternal protein malnutrition leads to abnormal insulin secretion in the next generation due to impaired liver function [44]. Many additional correlative studies have also shown that pre-conception paternal chronic ethanol treatment can result in increased alcohol sensitivity [45], increased anxiety-impulse-like behaviors [46], and diminished stressresponsivity [47] in offspring. Taken together with previous studies on human alcoholics showing differential miRNA expression in their brain and gut [48], we can speculate that spermatozoa sncRANs are also influenced by paternal environmental exposures.

Recently, mounting evidence has revealed that fathers can modify spermatozoa sncRANs’ expression under different environmental exposures. The spermatozoa sncRANs from Lean and Obese patients were found to be considerably different [49]. Furthermore, a mouse model of high-fat diet-induced obesity found that male obesity regulates spermatozoa microRNA’s abundance [50]. Furthermore, a high-fat diet down-regulates the expression of miRNA let-7c in the spermatozoa of rats and their offspring, which is involved in regulating glycolipid metabolism [38,51]. Following chronic ethanol exposure, Rompala et al. collected cauda spermatozoa for small RNA sequencing and discovered an altered abundance and post-transcriptional modifications. Notably, the expression of tRNA cytosine-5 methyltransferase, Nsun2, was reduced in male mice [52]. Furthermore, Nsun2 knockout mice showed an increase in tsRNAs [53]. This phenomenon of spermatozoa sncRNAs alteration has also been demonstrated in humans who were abused during childhood [54]. What is more, the spermatozoa sncRNAs’ profile is impacted in a sustained manner long after chronic stress exposure [55].

In-depth analyses of these altered spermatozoa sncRNAs downstream revealed that many are involved in early embryonic development and metabolic processes in the offspring. For example, stress-altered spermatozoa miR-155-5p facilitates mouse embryonic stem cells differentiation, miR-34c-5p triggers the first embryonic cleavage in mice, and miR-31-5p targets complement C1q Tumor Necrosis Factor-Related Protein 9A (CTRP9) protein involved in glucose metabolism [56]. To verify the effect of altered spermatozoa sncRNAs, Grandjean et al. injected miR-16b and miR-19, which are aberrantly expressed in the spermatozoa of mice fed high-fat diets, into naive embryos, generating offspring with obesity and type 2 diabetes phenotypes. Furthermore, these abnormal phenotypes can be recapitulated in offspring generated from embryos injected with diet-altered spermatozoa RNA [57]. Similarly, the advanced age of the paternal mice alters the spermatozoa tsRNAs profile. Male offspring generated by injecting spermatozoa tsRNAs from aged mice into zygotes show anxiety-like behaviors [58].

However, not all consequences of environmental-induced alterations in spermatozoa sncRNAs are harmful. Positive factors can prevent and offset the negative effects of paternal trauma on the offspring [59]. Previous studies have demonstrated increased levels of miR-193b expression in patients with type 2 diabetes [60]. However, spermatozoa miR-193b expression can be restored following exercise in fathers on high-fat diets, which has a favorable effect on insulin-mediated metabolic pathways in the offspring, reducing the risk of type 2 diabetes [61]. Similarly, the changes in some spermatozoa sncRNAs in the mouse model with four weeks of wheel running were opposite to those revealed by the long-term stress effect experiments. Furthermore, their male offspring exhibited significantly reduced fear and anxiety behaviors [59,62]. The KEGG pathway shows that some of these altered sncRNAs target genes are associated with receptors such as GnRH receptors and G-protein-coupled receptors, while others are involved in regulating synaptic transmission and other pathways [59]. It alludes that paternal exercise may protect against adverse environmental stress-induced changes in the expression of sncRNAs and their target genes. Remarkably, there is notable sexual dimorphism in the offspring phenotypes of exercising fathers. Fear memory does not change appreciably in female offspring [59]. Furthermore, weight gain in female offspring at weaning does not show corresponding changes in male offspring [61].

Paternal environmental exposures can affect his spermatozoa sncRNAs profile, which may be involved in regulating metabolic pathways, stress responses, and behavioral performance in the resulting progenies. More striking results implied that altered sncRNAs may modulate gamete information in the following generation, resulting in transgenerational inheritance. In conclusion, studies related to spermatozoa sncRNAs as carriers of intergenerational transmission of epigenetic information are clearly anincreasingly popular research area. Increased knowledge regarding these mechanisms by epidymosomal sncRNAs may help us find ways to optimize the survival of offspring by modifying the parental exposure environment (Table 1).

### 4.3. The Contribution of Epididymosomes to the Alteration of Spermatozoa sncRNAs

The new and historical data point to significant differences in spermatozoa sncRNAs between the cauda and caput epididymis [64]. A plausible explanation for the decrease in sncRNAs is that these sncRNAs are encapsulated in cytoplasmic droplets and are discarded as the sperm mature. However, the question of how spermatozoa acquire sncRNAs has not been precisely determined. Since spermatozoa are transcriptionally and translationally inert and unlikely to produce precursors for sncRNAs, this process must be driven by the complex epididymal microenvironment. Epididymosomes, thought to be key components in the microenvironment [3], are at the forefront of the candidates mediating this transformation.

Labeling using epididymis-specific expression of uracil phosphoribosyltransferase (UPRT) reveals that sncRNAs carried by mature spermatozoa in the cauda epididymis are first synthesized in the epididymal epithelium [65], supporting that the additional sncRNAs in spermatozoa originate from the epididymis. Moreover, the sncRNAs of spermatozoa and epididymosomes collected from the same epididymal region have substantial overlap and closely related abundance [55]. Researchers used corticosterone to treat mouse caput epididymal epithelial cells in vitro and then compared the miRNAs profile of the generated epidymosomes with those identified in spermatozoa recovered from stress. They identified a pattern of increasing miRNAs overlap over time following stress treatment [55]. Similarly, the tsRNAs in epididymosomes from mice chronically exposed to ethanol also showed equivalent alterations to those observed in their spermatozoa [52]. The consistency of these changes all provides evidence that epididymosomes may act as the primary information carriers that mediate changes in spermatozoa sncRNAs. Indeed, sncRNAs such as miR-375 and tRF-Gly-GCC carried by epididymosomes have also emerged in spermatozoa following co-culture in vitro [38,66].

Chan et al. extracted and purified caput epididymal spermatozoa and divided them into two groups that were incubated with either vehicle-treated or corticosterone-treated epididymosomes to generate zygotes via ICSI. As a result, abnormal neurodevelopment during the embryonic period was observed in the offspring of the corticosterone-treated group, particularly significant changes in genes related to synaptic signaling and neurotransmitter transport. Furthermore, the placental transcriptome was also affected, resulting in the enrichment of inflammatory and immune responses and a reduction in the genome associated with chromosomal and chromatin processes. Moreover, they could recapitulate the hypothalamic–pituitary–adrenal axis dysregulation shown in offspring from natural mating in adulthood [55]. This suggests that sncRNAs delivered by epididymosomes have a significant role in embryonic development and offspring health.

In addition, ejaculated semen also contains epididymosomes, which carry contents such as sncRNAs similar to mature spermatozoa and may also transmit information to the zygotes. Thus, increased knowledge of the potential new functions of the epididymosomes will not only provide an explanation for sperm dysfunction in infertile men, but will also shed light on the mechanisms by which paternal environmental exposure affects the postnatal phenotype.

## 5. The Possible Mechanisms by Which Spermatozoa sncRNAs Affect Embryonic Development and Offspring Health

It has been observed that environment-induced phenotypes are recapitulated in the progenies [67]. However, there is still a massive gap in knowledge about how spermatozoa sncRNAs carrying “environmental memory” can evade embryonic reprogramming to influence the next generation and even cause intergenerational inheritance. Based on existing studies, it is conjectured that once spermatozoa sncRNAs are delivered into the zygote, some shared regulatory mechanisms that govern embryonic development and offspring health are influenced early on by modifying genetic and epigenetic information. 

Regarding the role of spermatozoa sncRNAs in the offspring from embryo to adulthood, a “butterfly effect”, which is a transcriptional cascade effect, has been postulated [63]. A great amount of experimental evidence suggests that sncRNAs are importantelements in regulating gene expression, including controlling chromatin structure, regulating RNA transcription and modification, as well as influencing protein translation efficiency and stability [31,68,69,70]. Chen et al. analyzed sequence matches throughout the genome and found that spermatozoa tsRNAs from male rats on high-fat and normal diets exhibited distinct differences, with some preferentially matching gene promoter regions. Further analysis showed that the downstream pathways of these dysregulated genes include apoptosis, glucose metabolism, oxidative stress, and others [63]. Similarly, in eight-cell embryos and blastocysts, these tsRNAs also show differences [63]. These results suggest that environment-altered spermatozoa sncRNAs can be delivered into the zygote along with the paternal genome [23]. Furthermore, then, they have the potential to mediate the inheritance of paternal acquired traits by activating or repressing transcriptional start sites to regulate gene expression products in early embryos. Moreover, after being delivered into the oocyte, spermatozoa sncRNAs may modulate the stability and translational efficiency of maternal transcripts [23]. These changes will continue evolving along the female transcriptome and proteome to render further responses.

Some sncRNAs can bind directly to DNA methyltransferase (Dnmt), causing aberrant DNA methylation patterns [71]. For example, in early mouse embryos, miR-29b can interfere with DNA methylation patterns by suppressing Dnmt3a/3b expression, ultimately resulting in impaired mouse embryonic development [72]. It provides new ideas to answer the question of how the environment affects offspring by altering paternal spermatozoa sncRNAs. That is, this sncRNAs’ code may somehow be translated into other epigenetic modifications, guaranteeing the transmission of environment-induced paternal phenotypes.

It has been found that 5-methylcytidine (m^5^C) and N2-methylguanosine (m^2^G) are significantly upregulated in spermatozoa tsRNAs modifications in fathers on high-fat diets [63]. The presence of m^5^C has been reported to contribute to tRNA stability [63] and be associated with Dnmt2-mediated transgenerational epigenetic inheritance [73]. Stress-altered spermatozoa miRNAs target a large number of the protein-coding genes related to intergenerationally and transgenerationally inherited DNA Methylation Regions (DMRs) [67]. Perhaps it is by altering methylation modifications, which can regulate gene expression, that the epigenetic information carried by spermatozoa sncRNAs can evade embryonic reprogramming to be inherited. Moreover, sncRNAs may also modify histones so as to alter embryonic chromatin structure [74]. It has been shown that miR-125b down-regulates histone H3 lysine-9 tri-methylation (H3K9me3), which leads to heterochromatin relaxation [75]. Heterochromatin can maintain genomic stability, participate in chromosome segregation and homologous chromosome pairing, as well as regulate gene expression. Taken together, DNA methylation, histone modifications, and sncRNAs may interact and interweave into a complex network of epigenetic regulation involved in a range of parental environment-induced changes in offspring phenotypes.

Adequate placental function is a critical determinant of optimal fetal growth, which is closely related to cognitive development throughout the embryonic period and metabolic processes after delivery [61]. Parental high-fat diets damage placental tissue, resulting in neovascularization and affecting inflammatory cytokines that increase the risk of type 2 diabetes in the offspring [76,77]. It is speculated that sncRNAs may act as mediators between paternal exposures and the function of placental tissue. For example, a low-protein diet significantly upregulates tRF-Gly-GCC in spermatozoa, suppressing the endogenous retroelement MERVL gene [38], a totipotent gene that regulates placental size and function, to regulate the growth and development of the offspring. Spermatozoa miR-193b is reduced after paternal exercise, which inhibits the expression of mRNAs such as Slc38a2 and TNF-α in the placenta. While TNF can stimulate sodium-coupled neutral amino acid transporter 2 (Slc38a2/SNAT2) in placental tissue [61], which supplies the fetus with amino acids required for growth, it has also been established that a low placental Slc38a2/SNAT2 expression level results in fetal growth restriction [78]. The above studies suggest that spermatozoa sncRNAs may affect the development and health of the offspring by regulating placental tissue growth, altering placental gene expression, and interfering with placental nutrient transport (Figure 1).

Nowadays, understanding the role of spermatozoa sncRNAs in zygotes is still a great challenge. Exploration into these areas is still ongoing and is at the forefront of reproductive research. It will not only help advance our knowledge of environmental influences on offspring, but it can also inform therapeutic intervention strategies, such as improving embryo quality and producing healthy offspring.

## 6. Prospects and Challenges

The use of ART such as in vitro fertilization (IVF) and ICSI has brought hope to infertility patients, but the results are not always satisfactory. Especially for some men with unexplained infertility, ART works poorly, and the health of the resulting offspring may be jeopardized. Recently, EVs have been tried for therapeutic use in several diseases. Epididymosomes, a subset of EVs, make a significant contribution to the transfer of epididymal cargoes to spermatozoa [7]. Therefore, based on the study of the mechanism, epididymosomes may become a new target to solve the problem of infertility. For example, invitro co-incubation of spermatozoa with epididymosomes may increase the success rate of in vitro fertilization and the survival rate of the resulting embryos.

As a key carrier of paternal epigenetic information, sncRNAs have been a hot research topic in the field of reproduction. An increasing amount of evidence suggests that epididymosomes play an important role in spermatozoa sncRNAs alterations [38,55,66]. The epididymal microenvironment can be influenced by male infertility factors such as vasectomy obstruction, ligation, and infection. Whether these factors can affect spermatozoa sncRNAs and become detrimental to embryonic development and offspring health remains unknown [79]. Therefore, it is an exciting prospect whether we can use epididymosomes to effectively remodel spermatozoa sncRNAs and apply them to issues such as improving embryonic development and preventing adverse effects of paternal exposures on the health of the offspring.

However, the isolation and purification of epididymosomes remain cumbersome due to the limitations of existing technology and equipment [3]. Furthermore, rather than an understanding of mechanisms, most of the conclusions that spermatozoa sncRNAs regulate embryonic development and offspring health are dependent on observations of the phenomenon. In particular, there is a long span of time from sperm epigenetic inheritance to embryonic development to individual growth and even health problems in adulthood. This process includes not only various developmental phases, but also several crucial physiological processes. Therefore, it is very challenging to study. Health issues in adulthood also involve systems beyond reproduction, such as the nervous system, metabolism and endocrinology, and the immune system. Interprofessional collaboration is also required for in-depth examinations of these challenges.

## Figures and Tables

**Figure 1 ijms-23-10851-f001:**
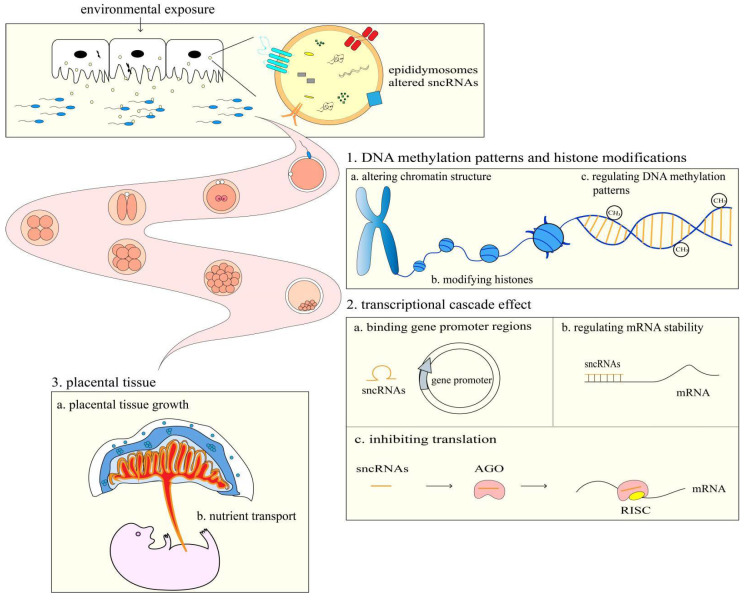
The possible mechanisms by which spermatozoa sncRNAs affect embryonic development and offspring health.

**Table 1 ijms-23-10851-t001:** Summary of recent studies on the role of spermatozoa sncRNAs in offspring health.

	Title	Author	Date	
Diet	Paternal obesity in a rodent model affects placental gene expression in a sex-specific manner	Binder et al.	2015	[42]
	Paternal obesity induces metabolic and sperm disturbances in male offspring that are exacerbated by their exposure to an “obesogenic” diet	Fullston et al.	2015	[43]
	High-fat diet reprograms the epigenome of rat spermatozoa and transgenerationally affects metabolism of the offspring	de Castro Barbosa et al.	2015	[51]
	RNA-mediated paternal heredity of diet-induced obesity and metabolic disorders	Grandjean et al.	2015	[57]
	Obesity and Bariatric Surgery Drive Epigenetic Variation of Spermatozoa in Humans	Donkin et al.	2016	[49]
	Biogenesis and function of tRNA fragments during sperm maturation and fertilization in mammals	Sharma et al.	2016	[38]
	Sperm tsRNAs contribute to intergenerational inheritance of an acquired metabolic disorder	Q. Chen et al.	2016	[63]
	Elevated paternal glucocorticoid exposure alters the small noncoding RNA profile in sperm and modifies anxiety and depressive phenotypes in the offspring	Short et al.	2016	[62]
	Epigenetic Mechanisms of Transmission of Metabolic Disease across Generations	Sales et al.	2017	[44]
Exercise	Exercise alters mouse sperm small noncoding RNAs and induces a transgenerational modification of male offspring conditioned fear and anxiety	Short et al.	2017	[59]
	Circulating ectosomes: Determination of angiogenic microRNAs in type 2 diabetes	Stępień et al.	2018	[60]
	Paternal high-fat diet and exercise regulate sperm miRNA and histone methylation to modify placental inflammation, nutrient transporter mRNA expression and fetal weight in a sex-dependent manner	Claycombe-Larson et al.	2020	[61]
Stress	Reduced levels of miRNAs 449 and 34 in sperm of mice and men exposed to early life stress	Dickson et al.	2018	[54]
	Reproductive tract extracellular vesicles are sufficient to transmit intergenerational stress and program neurodevelopment	Chan et al.	2020	[55]
	Early life stress affects the miRNA cargo of epididymal extracellular vesicles in mouse	Alshanbayeva et al.	2021	[56]
Ethanol	Paternal preconception ethanol exposure blunts hypothalamic–pituitary–adrenal axis responsivity and stress-induced excessive fluid intake in male mice	Rompala et al.	2016	[47]
	Heavy Chronic Intermittent Ethanol Exposure Alters Small Noncoding RNAs in Mouse Sperm and Epididymosomes	Rompala et al.	2018	[52]

## Data Availability

Not applicable.

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
