# Peer review of "sncRNAs in Epididymosomes: The Contribution to Embryonic Development and Offspring Health"

_ijms, 2022, doi:10.3390/ijms231810851_

Round 1

Reviewer 1 Report

This is an interesting review on a very relevant EV type for male reproduction, the epididymosomes. The authors have conducted an effective bibliographic revision and thorough discussion on the topic. Specifically, the authors focus on Epididymosome-mediated sncRNAs to the spermatozoa in the epididymal lumen. This study could be beneficial for researchers studying EVs and readers interested in reproduction broadly. I only have some minor comments.

Since the review is focused on sncRANs, I suggest modifying the title accordingly (Epididymosomes and sncRANs: the contribution to embryonic development and offspring health?).

This manuscript could be interesting for researchers not involved in molecular biology, I suggest defining sncRAN, tsRNA, snoRNA, etc. at first appearance (including the abstract). Indeed, some of them are defined, but after the first mention. Moreover, I believe IJMS encourages adding a table of abbreviations at the end of the manuscript. I suggest making one.

The language is correct, and the manuscript is easy to read and follow. Anyway, it might benefit from some simplification for clarity (e.g., L70-72).

In general, revise the text for unnecessary expressions. Especially the abstract. For instance, "It is generally believed." These kinds of expressions are cumbersome and reduce clarity while adding subjectivity. I suggest revising critically to remove them.

The abstract might add a couple of sentences summarising the most relevant information. It summarizes the current situation well but is not entirely informative.

Concerning the "gemmules" in the introduction, I would suggest using "hypotheses" instead of "theories." A theory in science is a well-established, robust, and evidence-supported body of thought (e.g., theory of evolution).

When discussing changes in RNAs "expression" (page 3), I suggest reminding that the use of that word does not imply that gene expression (in the canonical meaning) occurs in spermatozoa. L258 does this, but also elsewhere for clarity.

In L202 and elsewhere, I suggest including the species since the nomenclature could lead to confusion (e.g., here, "Lean and Obese patients" or "human males"). Adding the species add clarity to the text and removes the need to check the citation.

L241, is a citation lacking for sperm morphology?

L361, L341: Please, check punctuation ("and" after a dot).

L336-338: Please, check for clarity.

Are Figure 1 and Table 1 cited anywhere? All figures and tables should be cited in the text.

I suggest splitting the table into two pages for clarity when reading on PDF or paper, with their captions (maybe Fig. 1... continued; Fig. 1 (cont.)).

Also, I suggest making the figure as wide as the page (therefore, slightly enlarging the text). Changing the typeface to sans serif and making the text larger could benefit reading. Since it is preferable that figures do not interrupt text flow, moving L317-375 to the next page might aid the manuscript layout.

"Prospects and challenges" might need some citations. For instance, L391 about problems isolating epididymosomes.

L375: Revise if "while" and L375-377 are proper.

Reviewer 2 Report

We  are in presence of an epigenesis paper. Epigenesis involves Methylation of DNA and histones and miRNAs interference, in relation with chromatin (tertiary and biochemically) structure modification

First of all , several publications could be added to fit to the title of the paper

1-Altered sperm tsRNAs in aged male contribute to anxiety-like behavior in offspring; Yi --Guo, 1 , 2 Dandan Bai, 2 Wenqiang Liu, 2 Yingdong Liu, 

2-Small Noncoding RNAs Contribute to Sperm Oxidative Stress-Induced Programming of Behavioral and Metabolic Phenotypes in Offspring. Ren L, Xin Y, Sun X, Zhang Y, Chen Y, Liu S, He 

3-Changes in miRNA levels of sperm and small extracellular vesicles of seminal plasma are associated with transient scrotal heat stress in bulls. Alves MBR, Arruda RP, Batissaco L,

4-Exosomes of male reproduction. Baskaran S, Panner Selvam MK, Agarwal A

5-Sperm microRNA Content Is Altered in a Mouse Model of Male Obesity, but the Same Suite of microRNAs Are Not Altered in Offspring's Sperm. Fullston T, Ohlsson-Teague EM, Print CG, Sandeman LY, Lane M

Effect of freeze-thawing process on lipid peroxidation, miRNAs, ion channels, apoptosis and global DNA methylation in ram spermatozoa. Güngör BH, Tektemur A, Arkali G, Dayan Cinkara S,

Coding and Non-Coding RNAs, as Male Fertility and Infertility Biomarkers. Aliakbari F, Eshghifar N, Mirfakhraie R, Pourghorban P, Azizi F.

ncRNA BC1 influences translation in the oocyte. Aleshkina D, Iyyappan R, Lin CJ, Masek T, Pospisek M, Susor A.

Then, very few real proofs supporting the title/ paper. What we know is rather that these RNAs are able to regulate the first translation before Genomic activation/maternal to Zygotic transition.

It is known that ox stress modifies methylation; for the paternal genome there is an immediate demethylation followed immediately by a full re-methylation; the maternal demethylation is performed during cell division during the first preimplantation stages.

The oocyte is full of RNAse H (hybrid).  Difficult to conceive that they are stable all along the first stages of preimplantation development. Ref 2 links rather miRNAs to ox stress and abnormal  de-methylation. Literature is full of papers linking ox stress and methylation anomalies in reproduction: paternal age , diabetes… (Reprod Biomed Online. 2016, ;33(6):668-683, Fertil Steril. 2016;105(1):45-6 ; PLoS One. 2014 ;9(7):e100832.). When ox stress alters sperm, methylation modifications are transmitted to the next generation. Ref 5 confirms this is not the case for mi RNAs .

The paper is not convincing. miRNAS play a role in epigenesis and very early development. Not sure for the rest… to early , rather guessing. 

the success of intra testicular sperm retrieval for ICSI (see Cornell U data), in order to use the "best" spermatozoa is not in favor or the assertion. Moreover the transgenerationnal effect of ox stres via EDCs is strogly supported (See Manikkam et al. Plos one)

Reviewer 3 Report

The  role  of  epididymal extracellular vesicles  (EVs)  in sperm functions   has been   the   focus   of   several  studies. This review focused on the role and possible mechanisms of epididymosome-mediated transfer of small non-protein-coding regulatory RNAs (sncRNAs) in early embryonic development and subsequent health of the offspring. Evidence has shown that sncRNAs are implicated in a wide variety of cellular processes during epididymal maturation. It is a well-structured and organized Review. The Reviewer suggests that the following comments would be helpful to improve the quality of the paper.

1. The objective needs to be more explicit (L63-68). This Review is focused mainly on the role of epididymosome-related scnRNAs in early embryonic development and offspring health.

2. Please re-write. What are the several possible mechanisms regarding epididymosome-mediated communication with the sperm (91-92)?. So far only two possible mechanisms have been presented.

3. The possible mechanisms by which spermatozoa sncRNAs affect embryonic development and offspring health shown in Figure 1 are poorly explained. Please refer to the paper of Trigg et al. 2019 (Ref. #22) for more clarity on this matter.

4. Provide the full names of all abbrev/acronyms when used for the first time in the text.

5. Do the Authors have any published papers regarding the above-mentioned topics ((epididymosome-related sncRNA, sperm epididymal maturation, fertilization-associated processes, etc)? If so, please include them in the References. If not, how could you write a Review on the subject matter when you do not have any contributions to this field.

6. Please give the appropriate references for all the re-copied texts used in the Review. For examples,

a)"epididymosomes contribute to the sperm maturation process by mediating the crosstalk between the epididymis and the passing spermatozoa"- L15-16 (Zu et al., 2022, Biomedicines),

b) "The epididymis can be divided into three anatomical regions: the caput, the corpus and the cauda " - L73-74,

c) "MFGE8 integrin-binding Arg- Gly-Asp (RGD) tripeptide motif "- L109,

d) "generated zygotes via intracytoplasmic sperm injection (ICSI) using sperm obtained from the proximal (caput) versus distal (cauda) epididymis" -L135-136, and

e) "5-methylcytidine (m5C) and N2-methylguanosine (m2G) in sperm tsRNAs" -L330.

Round 2

Reviewer 3 Report

The Authors have addressed all of my comments, except for Comment #1 (from the previous review). The statements given in L65-67 are not the main objective of the study. This paper is focused mainly on the role  of epididymosome-related scnRNAs in early embryonic development and offspring health. Please see the statement in L19-21.
